# Trends and age-period-cohort analysis of migraine incidence in China from 1990 to 2021

**Yuting Huang** [*], **Dongxue Liu**[*], **Hongxiao Zhang**[‡], **Xufang Xu**[‡], **Shuangshuang Yuan, Shaoyang Cui, Zhihua Peng**[*], **Run Zhang**

Shenzhen Hospital (Futian) of Guangzhou University of Chinese Medicine, Shenzhen, China

☉ These authors contributed equally to this work.
‡ HZ and XX authors also contributed equally to this work.
* 490347656@qq.com

## Abstract

### Objective

The rapid economic development and demographic changes in Chinese society in the past 30 years may affect the prevalence of migraine. However, there are limited studies on the long-term trends and influencing factors of migraine burden in China. Using data from global burden of disease database (GBD2021), this study explores the changes in the migraine burden in China over a 30-year period and analyzes the main drivers.

### Methods

Joinpoint regression analysis was used to evaluate the temporal trend of migraine prevalence, incidence, and years lived with disability (YLDs), and the age-period-cohort (APC) model was used to evaluate the effects of age, time and birth cohort. Decomposition analysis were used to calculate the contribution of population growth, ageing, and epidemiological changes to the burden of disease.

### Results

In the past 30 years, the age-standardized prevalence rate (ASPR) and YLDs rate of migraine in China have increased significantly, and the disease burden of women is higher than that of men, especially in people aged 30–54 years. The main driver of the rising burden of disease is population growth. APC analysis showed that the younger generation had a higher burden of disease, which may be related to modern lifestyles and stressors.

### Conclusion

The burden of migraine continues to increase in China, which is mainly affected by population growth and epidemiological changes. In the future, attention should be

**Data availability statement:** All relevant data are within the paper and its Supporting Information files.

**Funding:** The author(s) received no specific funding for this work.

**Competing interests:** The authors have declared that no competing interests exist.

paid to high-risk groups, and lifestyle interventions and disease management should be strengthened to reduce the socio-economic burden.

## Introduction

Migraine is a very common neurovascular problem. It is characterized by repeated attacks, causing moderate to severe headaches, and recoverable neurological symptoms, such as photophobia, phonophobia, allodynia, nausea and dizziness. Almost one-third of patients will have temporary neurological phenomena, the most common of which is visual. These phenomena may occur before or during the headache, or independently of the headache [1]. Migraine often appears with psychological problems such as anxiety and depression, and it is also related to other physical problems, such as poor metabolism—especially insulin resistance and obesity [2]—and it is also closely related to cardiovascular diseases such as hypertension and stroke [3,4]. The range of associated comorbidities emphasizes that migraine transcends a purely neurological disorder, representing a systemic condition with significant clinical and public health relevance [5]. On a global scale, migraine is the second-ranked disabling disease, and it is the primary cause of female disability [6], but its prevalence varies greatly in different regions and populations [7]. Wider social and lifestyle changes-including more and more sedentary jobs, rising stress levels, changing eating habits to Western diet featuring high-sugar foods and processed foods, and increasing dependence on electronic devices-have been identified as potential triggers of migraine risk in previous studies. These factors may provide a meaningful background for us to understand the increasing global migraine burden, and point out the important direction of future empirical research [8]. The incidence of migraine, that is, the number of new cases, can more accurately reflect the change of risk over time than the total number of patients. Although this is very important for making preventive measures and policy planning, the continuous monitoring of migraine incidence in China is still insufficient. Many studies around the world have discussed the burden of migraine, few studies can make a detailed analysis according to the characteristics of these people, such as age, gender or birth year [7]. Moreover, the scope and methods of our existing Chinese research are very narrow, which makes it difficult for us to accurately judge the long-term development trend [9–11]. Many studies conflate incidence, prevalence, and YLDs, potentially undermining epidemiological clarity. Therefore, the present study seeks to fill these research gaps by analyzing migraine incidence, prevalence, and YLDs in China from 1990 to 2021, employing Joinpoint regression, age–period–cohort modeling, and decomposition analysis to improve interpretability across demographic groups.

## Methods

### Data sources

Our research uses the data of GBD 2021. GBD 2021 provides the latest estimates of diseases and injuries in 21 disease-burden regions and 204 countries and regions

from 1990 to 2021. All the data can be found publicly on the website of Global Health Data Exchange (https://vizhub.healthdata.org/gbd-results/). In GBD 2021, migraine was defined in accordance with the diagnostic criteria outlined in the *International Classification of Headache Disorders*, third edition (ICHD-3) [12], without differentiating between migraine with and without aura. For us in China, the input data comes from many places, including door-to-door population surveys, registration offices dedicated to recording certain diseases, and medical system databases, such as hospital records and medical insurance reimbursement lists [6,13]. The estimates were produced using the DisMod-MR 2.1 Bayesian meta-regression framework, which integrates heterogeneous data sources, enforces standardized case definitions, adjusts for recognized biases, and quantifies uncertainty through 95% uncertainty intervals [6]. This study utilized data on migraine-related indicators—prevalence, incidence, and YLDs—disaggregated by sex for China between 1990 and 2021. While modeled estimates enable extensive temporal analyses, discrepancies from observed diagnoses may arise due to heterogeneity in input sources and modeling parameters.

## Statistical analysis

**Descriptive analysis.** Data were organized and preprocessed in Microsoft Excel 2019. Summary statistics described the number of migraine cases and the age-standardized incidence, prevalence, and YLDs rates across different demographic groups. Visualization and graphical analyses were completed using R software (version 4.4.1).

**Joinpoint regression analysis.** Joinpoint regression was employed to assess temporal trends in the burden of migraine. The Annual Percent Change (APC), Average Annual Percent Change (AAPC), and their corresponding 95% Confidence Intervals (CI) were computed using Joinpoint software. An annual percent change > 0 was interpreted as an increasing trend, while an annual percent change < 0 was interpreted as a decreasing trend during the given time interval.

**Decomposition analysis.** We used a powerful analysis method in R software (version 4.4.1) to calculate how population growth, aging people and changes in disease patterns affected migraine burden from 1990 to 2021. As the previous study [14] said, this method can separate the influence of population structure from the influence of different ages and gender incidence changes. The total change in cases or rates ($\Delta Y$) was expressed as the aggregation of three components:

$$\Delta Y = M_p + M_a + M_m,$$

where $M_p$ represents the contribution of population growth, $M_a$ reflects the contribution of population ageing (i.e., changes in age structure), and $M_m$ captures epidemiological change, defined as shifts in age- and sex-specific incidence, prevalence, or YLDs rates after controlling for population size and age structure. The detailed mathematical formulation of the decomposition model, including the definition of interaction terms and their allocation, is provided in S1 File.

**Age-period-cohort analysis.** Age–Period–Cohort (APC) analysis in this study was conducted using the Biowinford online platform (http://biowinford.site:3838/trial/), which implements the constraint-based Holford identification method to resolve the intrinsic linear dependency among age, period, and cohort effects by imposing parameter constraints on period and cohort terms. Although the platform does not provide conventional diagnostics such as AIC, BIC, or residual analysis, this does not compromise methodological robustness, because likelihood-based information criteria are not applicable under APC's exact linear dependency structure—an established property documented in foundational APC methodological work [15]. The identification strategy used in Biowinford is identical to that of the NIH APC Web Tool (https://analysistools.cancer.gov/apc/), and the Holford estimable-function framework has undergone extensive validation across epidemiologic and demographic research. Empirical applications consistently demonstrate stable and reproducible age, period, and cohort trend estimates across large population datasets, supporting the reliability of APC results even in the absence of traditional diagnostics [16,17]. Accordingly, the consistency of our analytical framework with validated APC methodology, as well as the stable patterns observed in our estimable functions, supports the interpretability and robustness of the temporal trends reported in this study.

## Results

### Descriptive analysis

In 2021, the total number of migraine cases in China reached 184,752,280, comprising 7,003,623 males and 11,471,057 females (Table 1). Among these, 13,047,221 were new cases, with 4,965,958 in males and 8,081,262 in females. The total YLDs due to migraine amounted to 6,988,199, including 2,745,321 in males and 4,242,878 in females. The age-standardized prevalence rate of migraine was 11,777.5 per 100,000 population, with 8,782 in males and 14,959 in females. The incidence rate was 976 per 100,000 population, with 722.2 in males and 1,252.7 in females. The YLDs rate was 443.7 per 100,000 population, with 341.3 in males and 552.7 in females. Migraine was most common between ages 30–54, peaking in females aged 35–44. Age-standardized incidence began rising from 5–9 years and peaked at 30–39 years. The YLDs burden was highest in the 30–49 age group, with females experiencing a notably greater burden than males (Fig 1).

### Joinpoint regression analysis

Joinpoint regression analysis of age-standardized incidence (ASIR), prevalence (ASPR), and YLDs (ASYR) rates for migraine in China from 1990 to 2021 (Table 2) revealed dynamic trends. Both ASIR and ASPR declined from 1996–2000, with annual percent changes of −0.33% and −0.42%, respectively, increased sharply from 2000–2005 (1.00% and 1.19%), slowed during 2005–2016 (0.05% and 0.07%), accelerated in 2016–2019 (0.78% and 0.73%), and slightly changed in 2019–2021 (−0.16% and 0.12%). Gender-stratified analysis showed similar patterns, with females exhibiting marginally faster growth; male AAPC in incidence was 0.25% versus 0.18% in females, and overall ASPR AAPC was 0.23%. YLDs followed comparable temporal trends: decline (1996–2000), rise (2000–2005), plateau (2005–2016), acceleration (2016–2019), and slight decrease (2019–2021). Growth in YLDs was similar across genders (AAPC 0.24%), with females showing a slightly higher rate. These results indicate that migraine burden in China has fluctuated over three decades, with recent periods showing modest but consistent increases, particularly among females.

### Decomposition analysis

The decomposition analysis results (Fig 2) show that population growth was the dominant driver of the increasing burden of migraine in China. In both males and females, the contribution of population growth to incidence, prevalence, and YLDs was consistently large and nearly identical (Fig 2A–C). In contrast, the impact of population aging was relatively limited, exerting only modest effects on prevalence (Fig 2B) and YLDs (Fig 2C). Epidemiological change—defined as variation in age- and sex-specific incidence, prevalence, or YLDs after accounting for changes in population size and structure—also contributed to the observed increases. Its effect was moderate compared with demographic drivers, but was more

**Table 1. Migraine cases and age-standardized prevalence, incidence, and Years Lived with Disability rate in all age groups in China in 2021.**

| Measure | All-ages cases | | | Age-standardized rate per 100000 people | | |
|---|---|---|---|---|---|---|
| | Total | Male | Female | Total | Male | Female |
| Prevalence | 184752280 (160836525,213633958) | 70036223 (60652046,81324228) | 114716057 (98856231,132076638) | 11777.5 (10137.6,13538.6) | 8782.0 (7497.4,10129.3) | 14959.0 (12857.8,17190.4) |
| Incidence | 13047221 (11597731,14698852) | 4965958 (4347456,5638305) | 8081262 (7209929,9102490) | 975.6 (862.3,1102.1) | 722.2 (629.1,825.0) | 1252.7 (1112.5,1403.9) |
| YLDs (Years Lived with Disability) | 6988199 (1133319,15186289) | 2745321 (581677,5851443) | 4242878 (548620,9335915) | 443.7 (66.9,971.7) | 341.3 (67.7,739.2) | 552.7 (64.7,1221.5) |

Incidence: The occurrence of new migraine cases and captures temporal shifts in risk more directly than prevalence. Prevalence: The total number of existing cases at a specific time, reflecting the population burden. YLDs: Years lived with disability.

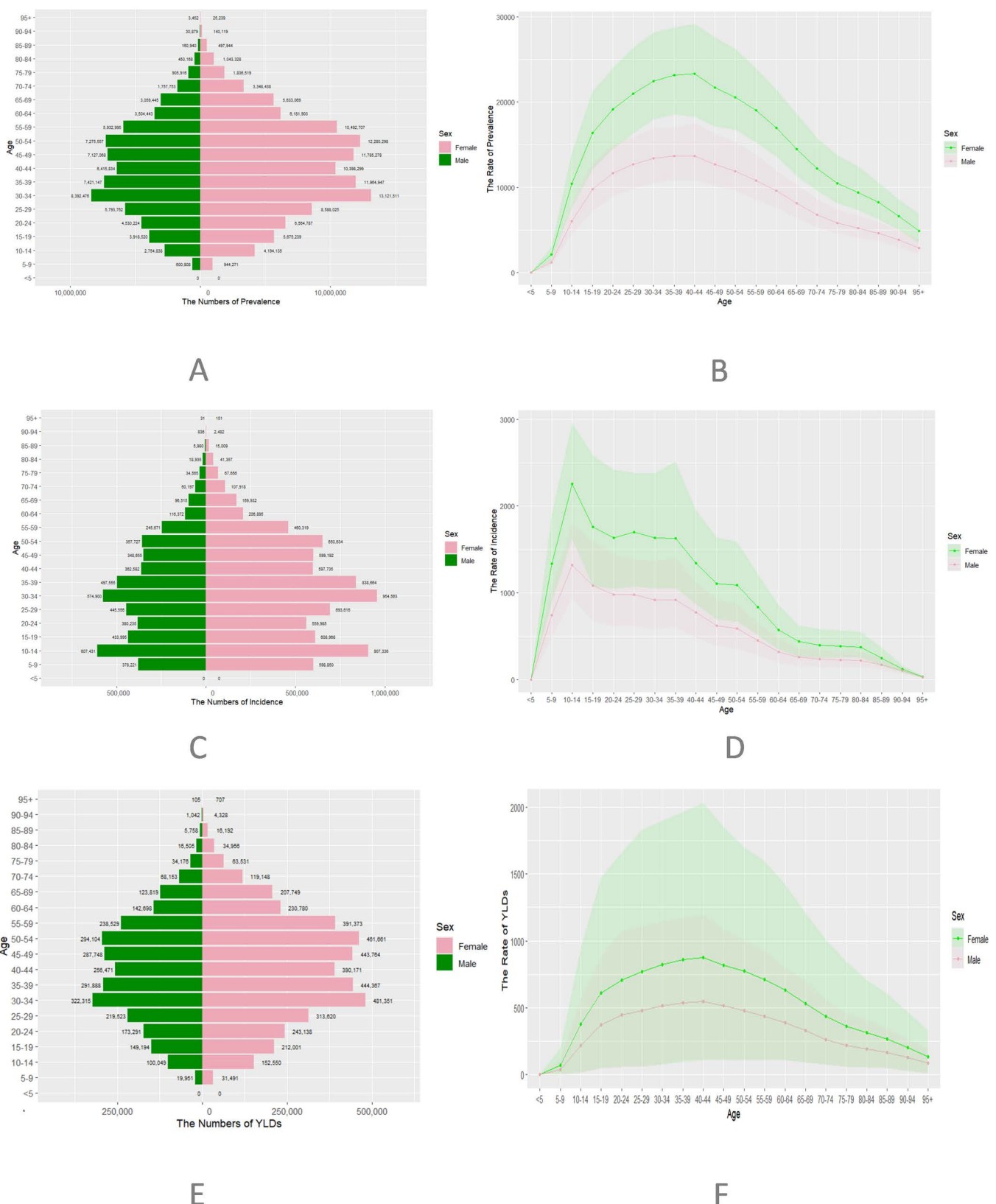

**Fig 1. Age-specific counts and age-standardized prevalence, incidence, and YLDs of migraine in China in 2021.** (A) Prevalence in specific age groups. (B) Age-standardized prevalence. (C) Incidence in specific age groups. (D) Age-standardized incidence. (E) YLDS in specific age groups. (F)

Age-standardized YLDS. incidence rose from early childhood (5-9 years), peaked in the 30-39 age group, and declined after age 60, with females consistently showing a higher burden than males.

**Table 2. Trends in the joint regression analysis of age-standardized incidence rates, prevalence rates, and years lived with disability rates for all genders, men and women in China from 1990 to 2021.**

| Gender | ASIR | | | ASRP | | | ASYR | | |
|---|---|---|---|---|---|---|---|---|---|
| | Period | APC(95%CI) | AAPC(95%CI) | Period | APC(95%CI) | AAPC(95%CI) | Period | APC(95%CI) | AAPC(95%CI) |
| Both | 1990-1996 | −0.03 (−0.08~−0.02) | 0.20 (0.16~0.24) | 1990-1996 | −0.02 (−0.08~0.04) | 0.23 (0.19~0.28) | 1990-1996 | 0.01 (−0.06~0.08) | 0.23 (0.18~0.29) |
| | 1996-2000 | −0.33 (−0.47~−0.19) | | 1996-2000 | −0.42 (−0.59~−0.25) | | 1996-2000 | −0.50 (−0.69~−0.30) | |
| | 2000-2005 | 1.00 (0.91~1.10) | | 2000-2005 | 1.19 (1.09~1.30) | | 2000-2005 | 1.19 (1.06~1.32) | |
| | 2005-2016 | 0.05 (0.03~0.08) | | 2005-2016 | 0.06 (0.04~0.09) | | 2005-2016 | 0.07 (0.04~0.10) | |
| | 2016-2019 | 0.78 (0.50~1.07) | | 2016-2019 | 0.87 (0.53~1.21) | | 2016-2019 | 0.73 (0.33~1.14) | |
| | 2019-2021 | −0.16 (−0.44~0.13) | | 2019-2021 | −0.07 (−0.42~0.28) | | 2019-2021 | 0.12 (−0.28~0.52) | |
| Male | 1990-1996 | −0.02 (−0.09~0.04) | 0.25 (0.20~0.30) | 1990-1996 | −0.02 (−0.06~0.03) | 0.27 (0.24~0.30) | 1990-1996 | 0.08 (−0.01~0.17) | 0.24 (0.20~0.29) |
| | 1996-2000 | −0.36 (−0.54~−0.18) | | 1996-2000 | −0.39 (−0.51~0.27) | | 1996-2000 | −0.49 (−0.61~−0.36) | |
| | 2000-2005 | 1.00 (0.89~1.12) | | 2000-2005 | 1.09 (1.01~1.17) | | 2000-2005 | 1.17 (1.05~1.30) | |
| | 2005-2016 | 0.10 (0.07~0.13) | | 2005-2016 | 0.04 (0.02~0.06) | | 2005-2016 | 0.20 (0.01~0.40) | |
| | 2016-2019 | 1.17 (0.79~1.56) | | 2016-2019 | 1.04 (0.91~1.17) | | 2016-2019 | 0.00 (−0.09~0.09) | |
| | 2019-2021 | −0.17 (−0.56~0.22) | | 2019-2021 | 0.02 (−0.23~0.28) | | 2019-2021 | 0.45 (0.38~0.52) | |
| Female | 1990-1996 | −0.02 (−0.06~0.01) | 0.18 (0.15~0.21) | 1990-1996 | −0.03 (−0.08~0.03) | 0.21 (0.17~0.26) | 1990-1996 | 0.00 (−0.06~0.05) | 0.24 (0.19~0.28) |
| | 1996-2000 | −0.34 (−0.45~0.24) | | 1996-2000 | −0.50 (−0.65~−0.35) | | 1996-2000 | −0.43 (−0.58~−0.27) | |
| | 2000-2005 | 0.99 (0.92~1.05) | | 2000-2005 | 1.22 (1.12~1.32) | | 2000-2005 | 1.03 (0.92~1.13) | |
| | 2005-2016 | 0.05 (0.03~0.07) | | 2005-2016 | 0.06 (0.04~0.09) | | 2005-2016 | 0.07 (0.04~0.09) | |
| | 2016-2019 | 0.62 (0.42~0.82) | | 2016-2019 | 0.68 (0.37~0.98) | | 2016-2019 | 1.12 (0.80~1.45) | |
| | 2019-2021 | −0.09 (−0.29~0.11) | | 2019-2021 | 0.02 (−0.29~0.33) | | 2019-2021 | 0.03 (−0.29~0.35) | |

APC: Annual Percentage Change; AAPC: Average Annual Percentage Change; ASIR: Age-Standardized Incidence Rate; ASPR: Age-Standardized Prevalence Rate; ASYR: Age-Standardized YLDs Rate.

noticeable in females, particularly for YLDs (Fig 2C). Overall, these results highlight that population growth is the primary determinant of migraine burden, while aging has only a minor effect, and epidemiological changes have contributed additionally, especially among women.

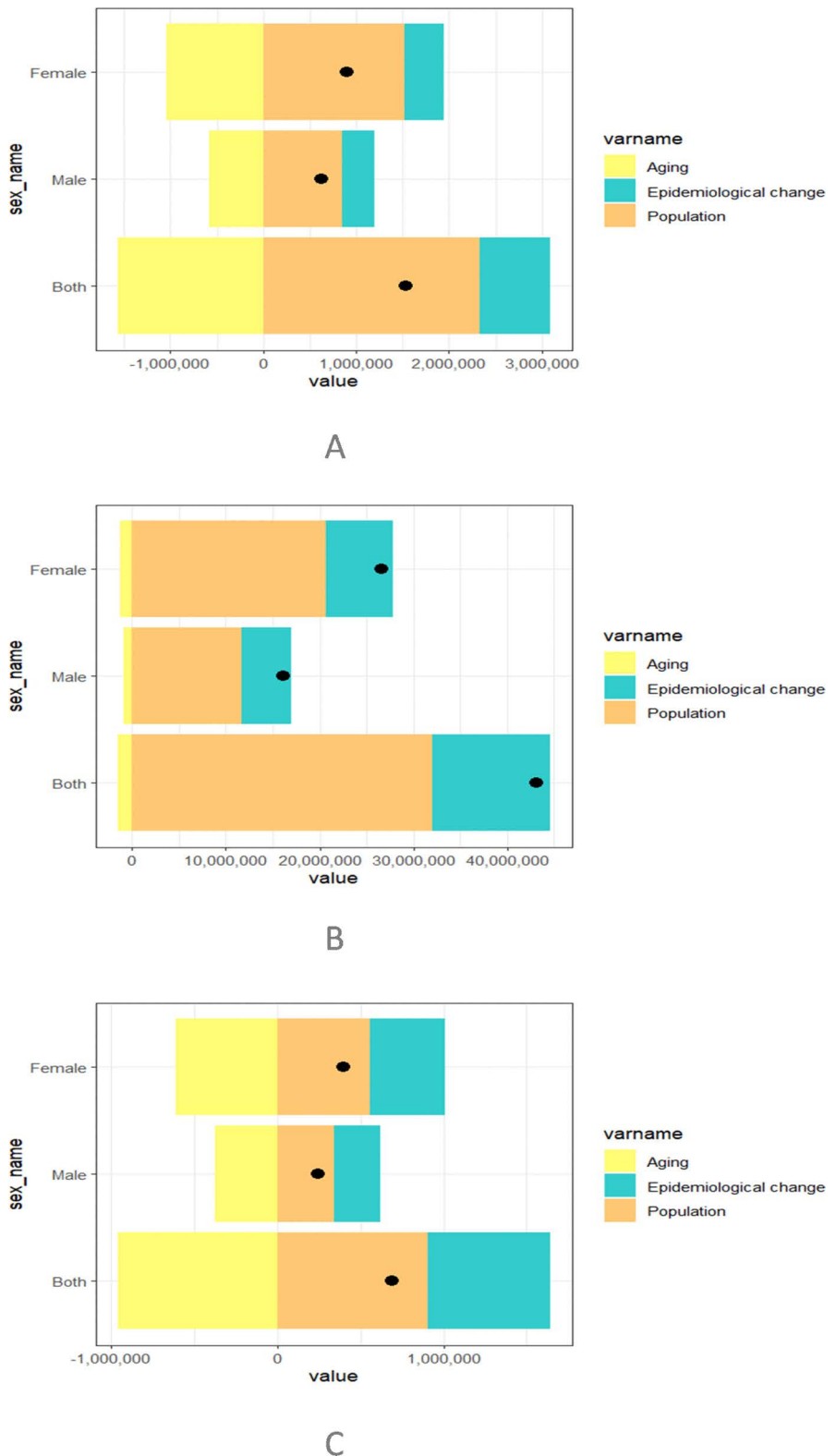

**Fig 2. Population level determinants of population growth, aging and epidemiological changes in Chinese from 1990 to 20221 on migraine incidence.** (A) Contribution of population growth, aging, and epidemiological change to migraine incidence. (B) Contribution of population growth, aging,

and epidemiological change to migraine prevalence. (C) Contribution of population growth, aging, and epidemiological change to YLDs rate. Population growth was the major driver of increasing incidence, while aging had limited impact. Epidemiological changes contributed modestly, with stronger effects observed among females.

### Age-period-cohort analysis

Age effect analysis reveals that the YLDs rate of migraine exhibits age-related variations (Fig 3, left column). The period effect reflects the temporal trend in migraine burden (Fig 3, middle column). According to the results of this study, between 1990 and 2005, the YLDs rate for migraine remained relatively stable; however, after 2005, an overall upward trend emerged, which became particularly pronounced after 2010. The period effect in females showed a significantly faster increase compared to males. The cohort effect reflects the changes in migraine burden across different birth cohorts (Fig 3, right column). The analysis indicates that more recently born cohorts (post-1990) exhibit an increasing trend in migraine YLDS compared to earlier birth cohorts.

### Discussion

People with migraine are more likely to encounter difficulties in life and their quality of life is more likely to deteriorate than those without headaches [18]. The burden brought by migraine is obviously different among people of different regions, sexes and levels of Socio-Demographic Index (SDI) [19]. These differences are closely related to population growth, aging, urbanization and lifestyle changes [20].

In order to find out the temporal trend of migraine burden, we analyzed the long-term data in different historical periods. From the mid-1990s to the early 2000s, the incidence and prevalence of migraine decreased slightly, which may be due to the problems of insufficient diagnosis and frequent misdiagnosis in primary health care, where the diagnostic accuracy was only 13%–18%, and people seldom used drugs such as triptans at that time [21]. Specific drugs for migraine, such as triptans, are few in number, which makes many people reluctant to see a doctor. Among all acute treatment drugs, the prescription of these specific drugs is less than 3.3%, and it is especially difficult to find them in rural clinics [22]. With our country's more and more understanding of migraine, the age-adjusted incidence and prevalence rate began to rise again between 2000 and 2005. This situation may be due to the popularization of clinical guidelines and the use of triptan therapy, which has brought some slow improvement [23], although direct supporting evidence remains limited. From 2016 to 2019, we see that the growth rate has become faster again, which is just the time when the "Healthy China 2030" initiative began to be implemented. The plan pays special attention to making more people enjoy medical services and increases the services of specialists. From 2019 to 2021, the data decreased slightly, which is probably due to the influence of COVID-19 epidemic. During that time, the number of people going to the hospital outpatient service decreased obviously. Although the number of people who see online doctors has increased, the accuracy of online diagnosis is not so high, so the data we see has decreased, which may not be that there are fewer migraine patients, but that some cases have not been recorded [24].

The results of this study tell us that the incidence of migraine begins to increase from the age of 5–9, reaches its peak at the age of 35–44, and then gradually decreases. The highest incidence rate is among people aged 30–54, and it will drop significantly after 60 years old. Therefore, migraine mainly bothers young people and middle-aged people. We also found a big difference between men and women. The burden of illness of women is much heavier than that of men [25]. These sex- and age-related variations in migraine burden can be explained by well-established neurobiological mechanisms [26].

Trigeminovascular pathway plays a central role in migraine attack. This pathway originated from the sensory neurons in the trigger ganglion, and their peripheral branches connected with intracranial blood vessels and dura mater, while the central part extended to the trigger nucleus caudalis in the brain stem. When these afferent fibers are activated, calcitonin gene-related peptide (CGRP) and other neuropeptides will be released in meningeal blood vessels, leading to vasodilation, plasma protein exudation and mast cell degranulation [26,27]. Together, these mechanisms inflame our nerves, so the pain signal spreads even more. In this way, the trigeminal nerve becomes particularly sensitive, so that those

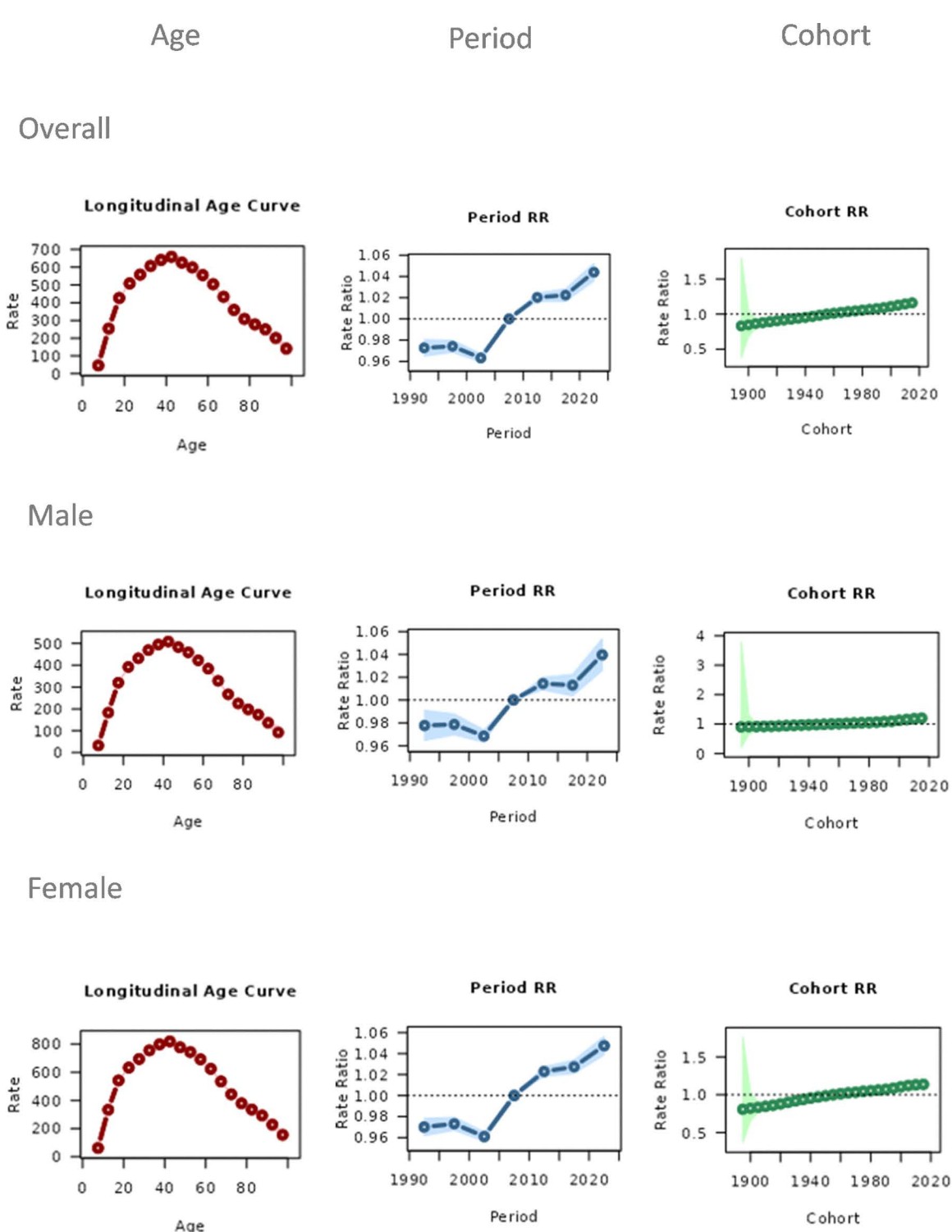

**Fig 3. Effects of age, period, and cohort on the rate of YLDs in migraine.** The age effect showed the heaviest burden in young and middle-aged adults, the period effect indicated rising YLDS after 2005, and the cohort effect revealed higher burdens in more recent birth cohorts, especially post-1990.

"pain signals" are constantly transmitted to the brain stem and cerebral cortex, so the headache can not be cured. Sex hormones-especially Estrogen-can also affect this process. Estrogen can affect CGRP signal in many ways, for example, it can promote gene transcription, help peptide release, and increase the number of receptor CLR and RAMP1 [28,29]. It also affects migraine-related neurotransmission by regulating serotonin and glutamate activity [12,30,31]. The changes in estrogen levels in our bodies can explain why migraines increase in adolescence, become the most severe in young adults, and then decrease after menopause. Monoclonal antibodies and small molecular antagonists that specifically deal with CGRP- trigeminal neurovascular pathway are really useful in clinic, which tells us that this pathway is super important in the process of migraine [32]. This effect can help us understand why migraine becomes significantly more common after puberty and why many women report that migraine is often related to menstruation. They also make us understand that the therapeutic effect will vary with gender, for example, women may have different reactions to CGRP-targeted interventions. Although females presently bear a higher migraine burden, demographic projections suggest that by 2050, the rates of increase in incidence, prevalence, and Disability-Adjusted Life Years (DALYs) among males will exceed those in females [33]. This may reflect higher work stress and limited health awareness among men [33], cultural barriers to reporting and treatment [34], and insufficient help-seeking [35], underscoring the need to reconsider migraine as a "female disease" [19] and to strengthen male-focused prevention and early intervention [7].

Another pivotal mechanism involved in migraine pathophysiology is cortical spreading depression (CSD), characterized by a slowly advancing wave of neuronal and glial depolarization followed by a sustained suppression of cortical activity. CSD is widely recognized as the electrophysiological basis underlying the migraine aura [36]. By disturbing the ion balance in our body, and letting some exciting signal substances such as glutamate and potassium escape, CSD will activate the pain receptors on the meninges through a system called trigeminovascular, thus establishing a connection between cerebral cortex overexcitation and headache onset [37]. Patients with migraine with aura may not only have special symptoms, such as blurred vision or abnormal sensation before headache, but also face higher long-term cerebrovascular risk. This is because CSD is thought to destroy blood-brain barrier and increase the risk of stroke [38,39]. This mechanistic framework explains why cortical hyperexcitability is not merely an epiphenomenon but an active contributor to both the phenotype and comorbid risks of migraine.

In addition to adult patterns, our findings revealed that migraine incidence can rise as early as 5–9 years of age. Consistent with this, Meng et al. reported that many cases actually begin in childhood or adolescence [40], with prevalence reaching 7–10% among school-aged children and up to 20% in adolescents [41]. However, children's migraine is often not found by doctors. Because it is often not the kind of headache that we usually know, but it will turn into a strange appearance such as stomachache, cyclic violence, or sudden dizziness [42]. Recognizing these atypical manifestations is crucial for achieving timely diagnosis and reducing the burden associated with inadequately treated childhood migraine. Headache disorders that emerge during adolescence may persist or even worsen over time, potentially developing into chronic and refractory forms that impose substantial burdens on both individuals and society [32]. Environmental triggers such as academic stress, sleep deprivation, and excessive electronic device use further exacerbate the risk of migraine onset in children and adolescents [43], with excessive device use identified as a significant factor [44]. Migraine is the leading cause of health loss between ages 15 and 49 [45], and onset typically occurs before 18, challenging the prior view of adult onset [46]. The age range of 15–39 is critical for education, career, and social development, with heightened susceptibility [47], while individuals aged 35–44 face dual career and family pressures contributing to peak prevalence and YLDs[7]. As global aging accelerates, children and adolescents will remain a key demographic, requiring targeted prevention and treatment to reduce social and economic burdens [40]. Young and middle-aged patients are also disproportionately affected by stigmatization, necessitating improved awareness, patient dignity, and policy interventions [48].

Baidu Index data shows that "how to quickly and effectively treat migraine" is the most searched migraine problem, which shows that patients are most concerned about how to control symptoms and treat themselves [49]. In China, the

condition of seeing a doctor and treating migraine is not good enough: only 26.4% of patients get acute drugs, 15% take preventive treatment, and about 40% never see a doctor [50]. This situation tells us that we urgently need to improve the diagnostic ability of doctors, make it easier for patients to get treatment, and strengthen everyone's health knowledge education.

With the increasing burden of migraine in China, it will bring more and more pressure to our medical system and social economy, which reminds us to find this problem early and deal with it well. It is very important to take targeted measures according to everyone's age, gender, SDI index and different symptoms, and to ensure that everyone can get medical services fairly [19]. ailor-made public health strategies can help alleviate chronic pain [47], while pain knowledge education through various channels can improve people's understanding and self-management ability [51]. It is very important to establish a comprehensive system integrating education, primary care, professional training and effect evaluation to realize long-term control [19].

## Limitations

Several limitations should be acknowledged in this study. First, all estimates were generated from the Global Burden of Disease (GBD) database, which, although incorporating Chinese data sources—surveys, registries, hospital records, and insurance claims—relies on statistical modeling rather than direct data analysis. As a result, the outcomes may not precisely represent primary national data, and biases arising from input quality or model specification could yield systematic error. Therefore, these results may not fully represent the real situation of our country, and if the quality of input data is not good or the model setting is problematic, systematic errors may occur. Second, some cases may not be correctly recorded or classified by doctors, especially in rural areas, and among men who are not convenient to see a doctor. Third, our analysis did not take into account the huge differences between different places, cities and rural areas, and between the rich and the poor in China. Because we did not study these differences carefully, the results may be biased. Fourthly, some important factors, such as the change of people's living habits, air pollution and urbanization, are not included in the analysis. Finally, the age–period–cohort model, while useful for exploring temporal dynamics, relies on assumptions that constrain causal interpretation. So when we look at our research results, we should remember these imperfections, which also shows that we need to test and expand our findings with more detailed clinical data from more people in the future.

## Conclusion

Drawing on data from the GBD2021 database, this study provides a systematic evaluation of long-term migraine burden trends in China. Across the past three decades, age-standardized prevalence and YLDs rates have increased markedly, especially among adults aged 30–54, with females experiencing a disproportionately higher impact. It is found that the increase of population is the main reason, and the change of disease pattern is the second reason, and the influence of population aging is relatively small. Age–period–Cohort analysis also found that the younger generation bears a heavier relative burden. Generally speaking, these findings tell us that it is necessary to formulate age and gender-sensitive measures for high-risk groups to alleviate the long-term health problems and socio-economic impact of migraine. Future research should continue to investigate the drivers of migraine burden more deeply and evaluate the effects of those targeted measures.

## Supporting information

**S1 File. The Decomposition Method.** This file provides the step-by-step description of the decomposition approach used to attribute changes in migraine incidence, prevalence, and YLDs to population growth, population ageing, and epidemiological change.
(DOCX)s

**S2 File. Raw Data.** This file contains the dataset used for calculating the incidence, prevalence, and YLDs of migraine in China, as well as the corresponding inputs for the decomposition and age-period-cohort analyses.
(ZIP)

## Acknowledgments

We thank the Institute for Health Metrics and Evaluation (IHME) for providing access to the Global Burden of Disease Study 2021 (GBD 2021) data, which formed the basis of our analyses. We also acknowledge all researchers and institutions involved in the collection, curation, and maintenance of the GBD 2021 dataset, whose efforts have enabled global health research such as the present study.

## Author contributions

**Investigation:** Xufang Xu.

**Methodology:** Shuangshuang Yuan.

**Software:** Dongxue Liu, Xufang Xu.

**Supervision:** Shaoyang Cui, Zhihua Peng.

**Visualization:** Dongxue Liu, Xufang Xu, Run Zhang.

**Writing – original draft:** Yuting Huang.

**Writing – review & editing:** Yuting Huang, Hongxiao Zhang.

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
