## [Decision Letter · Decision Letter 0]

6 Aug 2025

Dear Dr. Huang,

Thank you for submitting your manuscript to PLOS ONE. After careful consideration, we feel that it has merit but does not fully meet PLOS ONE’s publication criteria as it currently stands. Therefore, we invite you to submit a revised version of the manuscript that addresses the points raised during the review process.

We look forward to receiving your revised manuscript.

Kind regards,

Pengpeng Ye

Academic Editor

PLOS ONE

Journal Requirements:

3. We note that your Data Availability Statement is currently as follows: “All relevant data are within the manuscript and its Supporting Information files.”

Please confirm at this time whether or not your submission contains all raw data required to replicate the results of your study. Authors must share the “minimal data set” for their submission. PLOS defines the minimal data set to consist of the data required to replicate all study findings reported in the article, as well as related metadata and methods (https://journals.plos.org/plosone/s/data-availability#loc-minimal-data-set-definition ).

If your submission does not contain these data, please either upload them as Supporting Information files or deposit them to a stable, public repository and provide us with the relevant URLs, DOIs, or accession numbers. For a list of recommended repositories, please see https://journals.plos.org/plosone/s/recommended-repositories .

4. Please upload a copy of Figure 1 to which you refer in your text on page 5. If the figure is no longer to be included as part of the submission please remove all reference to it within the text.

5. Please upload a copy of Figure 2 to which you refer in your text on page 8. If the figure is no longer to be included as part of the submission please remove all reference to it within the text

6. Please upload a copy of Figure 3 to which you refer in your text on page 10. If the figure is no longer to be included as part of the submission please remove all reference to it within the text.

7. PLOS requires an ORCID iD for the corresponding author in Editorial Manager on papers submitted after December 6th, 2016. Please ensure that you have an ORCID iD and that it is validated in Editorial Manager. To do this, go to ‘Update my Information’ (in the upper left-hand corner of the main menu), and click on the Fetch/Validate link next to the ORCID field. This will take you to the ORCID site and allow you to create a new iD or authenticate a pre-existing iD in Editorial Manager.

Reviewers' comments:

Reviewer's Responses to Questions

**Comments to the Author**

1. Is the manuscript technically sound, and do the data support the conclusions?

Reviewer #1: Partly

Reviewer #2: Partly

2. Has the statistical analysis been performed appropriately and rigorously?

Reviewer #1: I Don't Know

Reviewer #2: I Don't Know

3. Have the authors made all data underlying the findings in their manuscript fully available?

Reviewer #1: Yes

Reviewer #2: No

4. Is the manuscript presented in an intelligible fashion and written in standard English?

Reviewer #1: Yes

Reviewer #2: Yes

Reviewer #1: As a clinical researcher specializing in headache medicine, I commend the authors for tackling a topic of global neurological and public health relevance. The analysis of migraine burden in China over a 30-year span using the GBD 2021 dataset, combined with Joinpoint and age-period-cohort (APC) methods, provides a valuable framework. However, several significant conceptual, methodological, and reporting issues need to be addressed to meet the scientific and editorial standards of PLOS ONE.

1. Strengths

Timely topic: migraine remains a leading cause of global disability, and long-term trend data from China are underrepresented in current literature.

Use of a robust, longitudinal database (GBD 2021).

Application of both Joinpoint regression and APC models enriches the temporal analysis.

Inclusion of gender-stratified findings enhances interpretability and clinical relevance.

2. Major Concerns and Required Revisions

A. Introduction Needs Strengthening

The introduction lacks critical context. It should:

Specify gaps in prior Chinese or GBD migraine trend analyses,

Clarify the motivation for focusing on incidence,

Briefly acknowledge limitations inherent in using modeled data.

Recommendation: Expand the introduction to articulate the scientific rationale and relevance of the research more clearly.

B. Terminological Confusion: Incidence, Prevalence, and YLDs

The manuscript frequently conflates incidence, prevalence, and YLDs. This undermines epidemiological clarity.

Recommendation: Clearly define each metric and apply terminology consistently.

C. Critical Omission: Definition of Migraine in GBD

The manuscript does not describe how migraine cases were defined in the GBD 2021 model. As migraine diagnosis is clinical and symptom-based (e.g., per ICHD-3), clarity is essential.

Recommendation: Specify:

Whether ICHD-2 or ICHD-3 was used,

The types of source data for China (e.g., surveys, claims),

How modeling handled case ascertainment and uncertainty,

The implications of using modeled incidence vs. observed diagnosis.

D. Insufficient Methodological Transparency in APC Modeling

There is no mention of how the APC identifiability problem was handled, nor are any diagnostics reported.

Recommendation: Provide:

Identification strategy (e.g., Holford, constraint-based),

Software used,

Model diagnostics (e.g., residual analysis, AIC/BIC, VIF).

E. Joinpoint Analysis Lacks Historical or Clinical Context

Timepoint changes (e.g., in 2000, 2005, 2016) are presented without connecting them to relevant clinical or policy developments in China.

Recommendation: Contextualize Joinpoint inflection years using historical changes in health policy, treatment availability (e.g., triptans, CGRP), or diagnostic practices.

F. Superficial Clinical Interpretation

Sex and age-related differences are discussed briefly, but without meaningful reference to pathophysiological mechanisms. Notably:

CGRP biology, trigeminovascular pathways, and cortical spreading depression are not mentioned.

The increase in incidence beginning at age 5–9 is not discussed in the context of pediatric migraine.

Recommendation: Enrich discussion with insights from headache neuroscience and pediatric migraine epidemiology.

G. Speculative Interpretations of Lifestyle Factors

Attributing trends to screen time, stress, and diet is speculative and not empirically tested.

Recommendation: Rephrase such points as hypotheses or broader societal factors, not study findings.

H. Missing Limitations Section

Most critically, the manuscript lacks any discussion of limitations, which is required by PLOS ONE.

Recommendation: Include a paragraph acknowledging:

Use of GBD-modeled data rather than clinical registries,

Potential for misclassification or underdiagnosis (especially in rural or male populations),

No geographic stratification within China,

Unmeasured confounders (e.g., urbanization, air pollution),

Constraints inherent in APC modeling.

3. Minor Comments

Language: Consider professional editing for grammar and sentence flow.

Figures: Simplify and ensure that legends clarify key trends.

Data sharing and ethical statements are appropriate.

Reviewer #2: Introduction:

1) Among migraine comorbidities, the authors should also mention metabolic conditions such as insulin resistance and obesity (Del Moro L, Rota E, Pirovano E, Rainero I. Migraine, Brain Glucose Metabolism and the "Neuroenergetic" Hypothesis: A Scoping Review. J Pain. 2022 Aug;23(8):1294–1317), as well as cardiovascular diseases (Guidetti D, Rota E, Morelli N, Immovilli P. Migraine and stroke: "vascular" comorbidity. Front Neurol. 2014 Oct 8;5:193; Bigal ME et al. Migraine and cardiovascular disease: a population-based study. Neurology. 2010 Feb 23;74(8):628–35).

2) I recommend replacing the phrase “high-fat and high-protein diets” with the more general term “Western diet”, as some studies have shown that ketogenic diets, which are high in fat, may actually reduce migraine frequency and severity. Conversely, growing evidence suggests that high-glycemic index foods are the main dietary contributors to increased migraine burden (Del Moro L, Rota E, Pirovano E, Rainero I. Migraine, Brain Glucose Metabolism and the "Neuroenergetic" Hypothesis: A Scoping Review. J Pain. 2022 Aug;23(8):1294–1317).

Methods:

3) Although the study provides insightful decomposition analyses illustrating the contribution of demographic and epidemiological factors to the migraine burden, the methodology is not explicitly described. No formal models, equations, or supplementary materials are provided to allow for replication or critical evaluation of the decomposition approach.

Results:

4) There is a discrepancy between the figures reported in the text and those in Table 1. For example, the text states “the total number of migraine cases in China was 18,475,280,” while Table 1 reports “184,752,280”.

5) Table 1 should be corrected: values should be reported with only one decimal place.

6) In the main text, epidemiological data should be rounded and reported without decimal places to improve clarity.

7) The concept of “epidemiological change” is vague and not adequately defined. The authors do not explain which variables were used to represent these changes or how they were measured.

Discussion:

8) The Limitations section should explicitly mention that the study relies entirely on modeled estimates from the Global Burden of Disease, without comparison to real-world national data (e.g., Chinese surveillance systems, hospital admissions, prescription databases). GBD estimates are derived from complex models and may overestimate or underestimate true values, especially in countries with limited high-quality local data. This could introduce uncontrolled systematic bias.

9) China is geographically and socioeconomically heterogeneous, but the study does not examine regional differences (e.g., by province), urban vs. rural settings, or socioeconomic strata. This is another potential source of bias that should be acknowledged.

10) The discussion section includes several topics that are only tangentially related to the study’s results, such as calcitonin gene-related peptide therapy, estrogen’s contribution, and the framing of migraine as a "female disease". These should be either removed or significantly reduced, as they are not directly supported by the findings of this study.

**Do you want your identity to be public for this peer review?** For information about this choice, including consent withdrawal, please see our Privacy Policy

Reviewer #1: No

Reviewer #2: **Yes:** Lorenzo Del Moro

---

## [Author Response · Author response to Decision Letter 1]

27 Oct 2025

We sincerely thank the reviewers and the editor for their insightful and constructive comments. We have carefully revised the manuscript in accordance with all the suggestions and provided a detailed point-by-point response in the attached document. We believe these revisions have substantially improved the quality and clarity of the manuscript. If there are any additional points that require further modification, we would be very pleased to address them.

---

## [Decision Letter · Decision Letter 1]

14 Nov 2025

Dear Dr. Huang,

Thank you for submitting your manuscript to PLOS ONE. After careful consideration, we feel that it has merit but does not fully meet PLOS ONE’s publication criteria as it currently stands. Therefore, we invite you to submit a revised version of the manuscript that addresses the points raised during the review process.

We look forward to receiving your revised manuscript.

Kind regards,

Pengpeng Ye

Academic Editor

PLOS ONE

Journal Requirements:

Reviewers' comments:

Reviewer's Responses to Questions

**Comments to the Author**

Reviewer #1: All comments have been addressed

Reviewer #2: All comments have been addressed

2. Is the manuscript technically sound, and do the data support the conclusions?

Reviewer #1: Yes

Reviewer #2: Yes

3. Has the statistical analysis been performed appropriately and rigorously?

Reviewer #1: Yes

Reviewer #2: I Don't Know

4. Have the authors made all data underlying the findings in their manuscript fully available?

Reviewer #1: Yes

Reviewer #2: Yes

5. Is the manuscript presented in an intelligible fashion and written in standard English?

Reviewer #1: Yes

Reviewer #2: Yes

Reviewer #1: Summary of Evaluation:

After reviewing the manuscript and the author's responses, the majority of the reviewer's concerns have been adequately addressed. The key changes have been effectively made, particularly in terms of clarifying the research gaps, the focus on incidence, and the modeling limitations. Below is my detailed evaluation:

Point-by-Point Comments:

1. Introduction Needs Strengthening (Point 1):

The revised introduction now clearly specifies the research gaps and the rationale for focusing on incidence rather than prevalence, and acknowledges the limitations of using modeled data.

Conclusion: Fully addressed.

2. Terminological Confusion (Point 2):

The definitions for incidence, prevalence, and YLDs are clearly provided, and the terminology is now applied consistently throughout the manuscript.

Conclusion: Fully addressed.

3. Definition of Migraine in GBD (Point 3):

The definition of migraine using ICHD-3 is now explicitly stated, along with details of the data sources for China (e.g., population surveys, registries, hospital records). The modeling approach (DisMod-MR 2.1) is also clearly explained.

Conclusion: Fully addressed.

4. Methodological Transparency in APC Modeling (Point 4):

The author has described the Holford method and the Biowinford platform used for APC modeling. However, the lack of diagnostics (e.g., AIC, BIC) is acknowledged but not fully explained.

Suggested Improvement: The author should further explain the impact of missing diagnostics on the robustness of the model and clarify how the Holford method is widely validated despite the lack of diagnostics.

Conclusion: Mostly addressed, but further clarification required.

5. Joinpoint Analysis Lacks Historical or Clinical Context (Point 5):

The author has successfully contextualized the Joinpoint inflection years with relevant historical and clinical factors, such as the Healthy China 2030 initiative and the availability of triptans.

Conclusion: Fully addressed.

6. Superficial Clinical Interpretation (Point 6):

Detailed explanations of CGRP, trigeminovascular pathways, and cortical spreading depression (CSD) are now included. The discussion of pediatric migraine and early onset is also well addressed.

Conclusion: Fully addressed.

7. Speculative Interpretations of Lifestyle Factors (Point 7):

The manuscript now correctly reframes lifestyle factors as potential hypotheses and societal factors rather than definitive findings.

Conclusion: Fully addressed.

8. Missing Limitations Section (Point 8):

A dedicated limitations section has been added, acknowledging key limitations related to the use of modeled data, potential misclassification, lack of geographic stratification, and APC model constraints.

Conclusion: Fully addressed.

9. Minor Comments (Point 9):

Language has been improved, figure legends are clearer, and data sharing and ethical statements are appropriate.

Conclusion: Fully addressed.

Reviewer #2: The authors have adequately addressed all the comments. I have no further recommendations before publication

**Do you want your identity to be public for this peer review?** For information about this choice, including consent withdrawal, please see our Privacy Policy

Reviewer #1: No

Reviewer #2: **Yes:** Lorenzo Del Moro

---

## [Author Response · Author response to Decision Letter 2]

25 Nov 2025

Dear Reviewers,

I would like to express my sincere gratitude for your thoughtful and constructive feedback on our manuscript. I have thoroughly revised the manuscript in response to all of your comments, making the necessary changes to enhance the clarity and rigor of the work. The revised sections are highlighted in the revised manuscript with track changes.I noticed that the clarity of Fig. 1 in the final generated PDF file is not satisfactory. I have tried many methods to improve the clarity of Fig. 1 in the PDF, but the results have been unsatisfactory. I have separately uploaded Fig. 1 in TIFF format to the submission system and kindly request that the editor and reviewers review the JPG version I uploaded. I also respectfully ask that the manuscript not be rejected solely because Fig. 1 does not appear clearly in the PDF. If there are any further modifications or clarifications needed, I would be more than happy to make additional revisions and cooperate fully to ensure that the manuscript meets the journal's standards.

Thank you again for your time and invaluable input.

Best regards,

Yuting,Huang

First Author

---

## [Decision Letter · Decision Letter 2]

30 Nov 2025

Trends and age-period-cohort analysis of migraine incidence in China from 1990 to 2021

PONE-D-25-16628R2

Dear Dr. Huang,

We’re pleased to inform you that your manuscript has been judged scientifically suitable for publication and will be formally accepted for publication once it meets all outstanding technical requirements.

Kind regards,

Pengpeng Ye

Academic Editor

PLOS ONE

Additional Editor Comments (optional):

Reviewers' comments:

Reviewer's Responses to Questions

**Comments to the Author**

Reviewer #1: All comments have been addressed

2. Is the manuscript technically sound, and do the data support the conclusions?

Reviewer #1: Yes

3. Has the statistical analysis been performed appropriately and rigorously?

Reviewer #1: Yes

4. Have the authors made all data underlying the findings in their manuscript fully available?

Reviewer #1: Yes

5. Is the manuscript presented in an intelligible fashion and written in standard English?

Reviewer #1: Yes

Reviewer #1: All of my previously addressed points are satisfactorily resolved.

No remaining mandatory revisions.

**Do you want your identity to be public for this peer review?** For information about this choice, including consent withdrawal, please see our Privacy Policy

Reviewer #1: No

---

## [Editor Report · Acceptance letter]

PONE-D-25-16628R2

PLOS One

Dear Dr. Huang,

I'm pleased to inform you that your manuscript has been deemed suitable for publication in PLOS One. Congratulations! Your manuscript is now being handed over to our production team.

Kind regards,

on behalf of

Dr. Pengpeng Ye

Academic Editor

PLOS One